# Maternal Reflective Functioning and Attachment Security in Girls and Boys: A Closer Look into the Middle Childhood

**DOI:** 10.3390/ijerph182111261

**Published:** 2021-10-27

**Authors:** Danguolė Čekuolienė, Lina Gervinskaitė-Paulaitienė, Izabelė Grauslienė, Asta Adler, Rasa Barkauskienė

**Affiliations:** Institute of Psychology, Vilnius University, Universiteto St.9/1, LT-01513 Vilnius, Lithuania; lina.gervinskaite@fsf.vu.lt (L.G.-P.); izabele.grausliene@fsf.vu.lt (I.G.); astazb@gmail.com (A.A.); rasa.barkauskiene@fsf.vu.lt (R.B.)

**Keywords:** attachment security, maternal reflective functioning, middle childhood, gender

## Abstract

Child attachment undergoes major changes during middle childhood. Maternal reflective functioning (RF) is hypothesized to be an important correlate of a child’s attachment security during this period; however, the child’s gender role in this association has not been examined yet. In the present study, we used 64 mother-child (6–11 years old) dyads from a community sample to analyze the association between maternal RF and child attachment security and whether this link is moderated by gender. Maternal RF was assessed on the Parent Development Interview Revised (PDI-R2) and child attachment classifications were examined by the Child Attachment Interview (CAI). Results revealed the positive and statistically significant association between maternal RF and child attachment security in the whole sample. Further evaluations of this link in the groups of girls and boys separately indicated its significance for girls only. Finally, moderation analysis demonstrated the relation between child attachment security and maternal RF to be moderated by gender. These findings provide a new knowledge on gender role in attachment security in relation to maternal RF as well as suggest possible differentiation in the correlates on the pathway of secure attachment between girls and boys during middle childhood.

## 1. Introduction

A substantial amount of research on attachment over the last three decades has explored attachment behavior during early childhood and attachment representations in adults and adolescents [1,2]. Only recently have researchers turned their focus to the period of middle childhood–an age marked by a lot of critical changes in development. Major efforts in the research and theory have concentrated on the analysis of changes in attachment representations to the mother and father [3] and gender-specific qualities of attachment relationship at this age [4]. Along with existing studies in the area, the present paper explores the attachment representations in the context of maternal reflective abilities and security of attachment in middle-childhood boys and girls.

Attachment theory regards internal working models (IWMs) of attachment relationships as cognitive/affective constructs which develop in the course of behavioral interactions between the infant/child and their parents [5,6,7,8]. Initially, patterns of attachment reflect the child’s expectations about the parent’s likely behavior in various stressful situations. Eventually, the child generalizes these expectations to a set of ideas about how close relationships are experienced and how they function in daily life [9]. The idea that caretaking circumstances contribute to different patterns of attachment between the child and his/her mother was supported and specified by numerous empirical findings in many cultures [10,11,12]. The *caregiver’s sensitivity and responsiveness* to the infant’s signals were identified as one of the most powerful determinants in the development of a secure relationship between them [13]. However, an older and more competent child will need the caregiver’s support in different ways than an infant [8,14]. Internal working models are theorized to mediate the child’s behavior with the caregiver, the child’s self-concept, and the child’s interpersonal relationships, including their later attachments with peers and partners [8,9,15]. Therefore, communicational characteristics between the child and his or her parents ought to be taken into account while investigating attachment representations in older children [7].

### 1.1. Attachment in Middle Childhood: Elaborating on Communication and Reflective Dialogues between Children and Parents

During the period of early development, parents remain the primary attachment figures and the most important people in the social environment. Bowlby and Ainsworth suggested that the goal of the attachment system changes from the *proximity* of the attachment figure in early childhood to the *availability* of the attachment figure in middle childhood. The fourth phase of attachment–the goal-corrected partnerships [5,6,16]–assumes a child’s more developed understanding of their parent’s desires, communications, and decisions. Verbal information from parents can clarify a child’s nonverbal experience, but also may be at odds with it [7]. Thus, certain maternal verbal characteristics and communication competence were spelled out as important components in mother-child relationships. Fonagy, Steele, Steele, Moran, and Higgit [17] fostering this direction of attachment theory and research has turned attention to the *internal* qualities that allow mothers to be sensitive (i.e., *external* behavioral quality). They have focused upon a maternal individual capacity especially important during interactions in close relationships, i.e., “reflective functioning” (RF)–a mother’s capacity to understand that her own or other’s behaviors are linked in a meaningful, predictable way to underlying mental states such as feelings, wishes, thoughts, and desires [17]. In other words, RF refers to the awareness that an individual’s behavior is a reflection of unobservable internal experiences: investigation of the role of parental reflective functioning in children’s attachment formation marked a new turn in this domain of research [18,19]. Now, it has been more than three decades since parental reflective functioning as a possible precursor of children’s attachment security attracted scholars’ interests in the field of research of children’s attachment representations [20,21,22].

Another important growing point in empirical research of attachment was marked by increasing attention towards the exploration of different correlates and manifestations of attachment representations in older age groups, primarily adults and adolescents. Middle childhood, for some time, appeared to be a little neglected in this respect, however. Recently, researchers have turned their interests towards the multiple characteristics of the link between parenting and parent-child attachment qualities in this developmental stage [2,23].

Middle childhood is a time when children’s social world expands. School demands and communicating with peers begin to be more and more important. Advances in cognitive development (including self-cognition) permeate all areas of their lives as well as close relationships in the family. Thus, caregivers’ verbal competencies and quality of dialogues with their children not only become a significant component of parent-child relationships but may also serve as a rich source of information about the quality of attachment between them.

In spite of growing body of data about attachment in middle childhood, certain characteristics of parent-child communication that may be associated with the child’s attachment (in)security in middle childhood seems to be still under-examined.

The communication perspective on attachment relationships is based on the assumption that parents convey their own patterns of relating to children initially through behavioral-affective interaction patterns, but later also through verbal dialogue about past, future, and hypothetical experiences [14]. Importantly, it is suggested that how children construct their working models through interaction and how they learn to share them with parents and others through dialogue not only *reflect* but also *create* their relational realities [8]. Thus, parent-child conversations about the child’s experiences may significantly influence children’s understanding of events and in shaping their expectations of the caregiver [9,24,25]. As such, along with other important variables, parental verbal skills may play an important role in the process of construction and reconstruction of attachment relationships, especially when a child’s verbal efficiency expands. Research on mothers reminiscing style revealed that mothers of securely attached children are more elaborative and evaluative than are mothers of insecurely attached children [26,27,28]. It has been suggested that this may help children to develop richer and more coherent (i.e., more secure) internal working models of attachment [29].

Building out on this line of investigation allows for the possible and considerable overlap between the concept of the mother’s elaborative style in communication with the child and the construct of maternal reflective functioning. In this perspective, exploring the links between maternal reflective functioning and the child’s attachment security in middle childhood may suggest a new amplification of empirical knowledge of attachment representations at this age.

An especially intriguing question in the context of parental communication with children about interpersonal experiences is about the role of a child’s gender in these conversations. Do parents and specifically mothers take different styles in talking emotions to their daughters and sons? Some studies in the field of emotional dialogues between parents and children indicate that emotional content of mother-child conversations about the past is related to gender: findings indicate that mothers are more expressive and elaborative with daughters than with sons, especially about emotions [30,31,32]. All this allows us to suggest that different styles of maternal emotional experience conversations with sons and daughters may be associated with more coherent narratives on attachment related events in girls during the middle childhood–time when the importance of verbal communication between parent and child becomes more significant.

Then again, in the recently accomplished meta-analytic study on gender comparisons in mother-child emotion talk, Aznar and Tenenbaum [33] found no gender differences in the frequency of emotion talk between mothers of daughters and mothers of sons. This finding contrasted with findings mentioned above that mothers tend to be more emotionally expressive with their daughters than with sons [30]. Nevertheless, authors agree that only *frequency* and no other characteristics of mothers’ emotional conversations were taken into account in their investigation [33]. Elaboration, content, and valence involved in mothers’ emotional talk to their children may lead to a more complex style of communication overall. How may this style reflect on their relationships and, in turn, on differences in the security of girls and boys later? The ongoing theoretical dispute and inconclusive empirical findings in the literature on attachment suggests some controversy about the gender role in security of attachment during middle childhood [4]. Thus, further research in this direction is undoubtedly important.

To summarize, three main points in middle childhood attachment development should be considered in further research: (1) a shift to the centrality of language as a means of communicating emotions, needs, and relationships between parents and children; (2) specificity and centrality of attachment to mothers and fathers as figures in the development and generalization of attachment representations; and (3) gender-specific patterns of socialization of emotions and interpersonal experience conversational style with of boys and girls.

### 1.2. Current Study

The main goals of the present study are: (1) to investigate *maternal reflective functioning* as an important factor contributing to children’s attachment quality in middle childhood; (2) to examine *gender-related aspects* in the association between maternal reflective functioning and attachment security in middle-childhood boys and girls.

Presented disputes in literature on empirical findings, as well as suggested in this paper, overlap between the concept of mother’s elaborative style in communication with children about emotions and the construct of maternal reflective functioning allow us to assume the greater possibility for maternal reflective function to be associated with attachment security in girls.

To our knowledge, research to this day has not yet analyzed this direction of complex interlinks among these variables as well as possible moderating effects of gender in the relation of mothers’ reflective functioning abilities and children’s attachment security.

## 2. Materials and Methods

### 2.1. Participants and Procedure

Sixty-four mother-child dyads participated in this study. The sample included 33 boys (51.6%) and 31 girls (48.4%) aged from 6 to 10 (SD = 1.09, mean = 8.59). The mean age of mothers was 38.02 years (SD = 6.79, range 28–56). All participants were from the community sample. Forty-six children (71.9%) lived in families with either biological parents or stepparents, eight children (12.5%) lived in divorced families, six (9.4%) children were living in the single-parent family, and two (3.1%) children were in foster care. The parents of two (3.1%) children did not provide information about their family situation.

The sample of 64 dyads of children and their mothers was formed from a larger number of participants in a scientific research project on attachment and psychosocial development in middle childhood approved by the Lithuanian Research Council. The community sample for this project was recruited from schools in one of the Lithuanian regions. With the permission of school administrators, all students in grades 1–3 were asked to pass on a written invitation to their parents to participate in the study along with full information about the research, questionnaires, and informed consent. Parents were asked to provide their contact information if they agreed to be contacted personally for research purposes. From 451 parents who agreed to participate in the study, 292 (64.75%) parents provided their contact information. An attempt was made to contact 275 (94.2%) and invite them to participate in the interviews: mothers–the Parent Development Interview [34], children–the Child Attachment Interview [35]. One hundred and twenty two (44.4%) parents gave consent for their children to take part in the Child Attachment Interview, while 64 women agreed to take part in the Parent Development Interview. The interviews were administered by the trained researchers from the research project group.

The current study was conducted in compliance with the guidelines reported in the Declaration of Helsinki. Specifically, both parents and children were assured that their participation in the interviews was entirely voluntary and they were free to decline the process of the interview at any point. The study followed the ethical guidelines respecting respondents’ confidentiality for both data collection and data analysis.

### 2.2. Measures

**The Child Attachment Interview** (CAI, [35]) was used to assess attachment organization and security in middle childhood based on attachment representations. The CAI is a semi-structured interview, in which children are invited to describe their relationships with their primary caregivers. The interview consists of 19 questions that include themes about relationships, times of conflicts with caregivers, hurt, illness, distress, separation, and loss. It takes between 30 and 60 min to complete. The CAI coding and classification system comprises of nine subscales which are rated according to the manual from 1 (the lowest score) to 9 (the highest score): Emotional Openness, Balance of Positive/Negative References to Attachment Figures, Use of Examples, Resolution of Conflicts, Preoccupied Anger, Idealization, Dismissal, and Overall Coherence with Disorganization rated dichotomically as absent or present. The CAI has separate Preoccupied Anger, Idealization, and Dismissal subscales for mother and father, and these are coded separately.

The CAI coding and classification system is designed in a way that on the basis of all nine subscales, four attachment classifications can be assigned: Secure, Dismissing, Preoccupied, and Disorganized. The coding system also allows two-way classification, where Dismissing, Preoccupied, and Disorganized categories are combined into an Insecure attachment classification. The CAI was coded by two trained and certified coders, who had completed the reliability training and certification process. As mentioned above, this study was a part of a broader research project. In this project, 33.6% of all 122 interviews were randomly selected and double coded for reliability by two coders. For the two-way classification (secure versus insecure), interrater agreement was 91.8% (Kappa = 0.83, *p* < 0.001, *n* = 49) with regard to attachment with mother and 88.6% (Kappa = 0.76, *p* < 0.001, *n* = 44) with regard to attachment with father. For three-way and four-way classification, there was substantial interrater agreement, which ranged from 81.8% (Kappa = 0.69, *p* < 0.001, *n* = 44) to 87.8% (Kappa = 0.79, *p* < 0.001, *n* = 49). Intraclass correlation coefficients (ICC) between coders were substantial for three subscales ranging from 0.78 to 0.79 and almost perfect for eight subscales ranging from 0.81 (Anger with Father) to 0.96 (Use of Examples). However, the ICC of one subscale (Emotional openness) was fair (0.34). Nevertheless, the median ICC indicates a very strong agreement between the two coders–for all scales it was 0.84.

**The Parent Development Interview-Revised** (PDI-R2, [34]) was used for coding maternal reflective function (RF) from it. The PDI is used to evaluate parents’ representation of their children between 2 and 16 years old. The PDI is a semi-structured interview and consists of 40 questions. Questions are grouped into seven sections: parent’s view of the child, parent’s view of their relationship with the child, emotional parenting experience, mother/father family history, dependence/independence, separation/loss, and a look at the past and a look to the future. The PDI asks the parents to describe their current relationship with the child by providing examples of their everyday life. The interview takes about 1.5–2 h to complete. The PDI coding system has an 11-point RF rating scale, where one negative value (−1) is given, therefore that is from −1 (avoidance or active refusal to mentalize–negative reflective function) to 9 points (exceptionally rich, complete, and sophisticated understanding of mental states in interaction) with a mean value of 5 (clear or average reflective function, when the parent is able to give meaning to thoughts and feelings according to experience, but it is difficult to regulate more complex experience, for example, conflicts and ambivalence). The main types of reflective function indicators are: (1) an awareness of the nature of mental states, (2) the explicit effort to tease out mental states underlying behavior, (3) recognizing developmental aspects of mental states, and (4) mental states in relation to the interviewer. After the interview is transcribed, the certified coders of the RF on the PDI-R2-S scored mothers’ reflective function. Two coders, who had completed the reliability training, independently coded 23.4% (from 64 interviews, *n* = 15) randomly selected interviews. The intraclass correlation (ICC) showed good interrater agreement (r = 0.93, *p* < 0.001).

### 2.3. Data Analysis

Data analyses were conducted in SPSS Version 27 (IBM Corp, New York, NY, USA, 2020). First, we computed the descriptive statistics and examined relations between demographic variables and measures of child attachment classification (Secure vs. Insecure) using one-way analyses of variance (ANOVAs) and Chi-square tests. Correlation analysis to test the possible demographic covariates for the maternal reflective functioning was conducted. Next, to evaluate associations between maternal reflective function and child attachment, Spearman correlations (for attachment classification variable as well as CAI scales with not normal distribution) and Pearson correlations (for CAI scales corresponding to normal distribution) were calculated in the entire sample, as well as in girls’ and boys’ groups separately. Further, gender differences in continuous attachment variables (CAI subscales) and maternal RF were analyzed using Student t or Mann-Whitney tests. Finally, the proposed moderation model was examined using the PROCESS Macro v.3.4 for SPSS. Relevant to the current study, for dichotomous outcome variables PROCESS employs logistic regression, and Model 1 [36] with 5000 bias-corrected bootstrap samples was chosen. Through bootstrapping, the distribution of effects was empirically approximated and used for calculating confidence intervals. Index of moderation whose 95% lower and upper confidence intervals (95% BCaCIs) did not include 0 was considered statistically significant [36].

## 3. Results

### 3.1. Descriptive Statistics and Identification of Covariates

The distribution of attachment classifications showed that more than a half of children of the sample were classified as secure with respect to their mother and father (64.1% and 57.8%, respectively); the classification of dismissing attachment was detected in 23.4% of children with their mother and 20.3% of children with their father. The frequency of disorganized attachment was 12.5% for the mother and 10.9% for the father. There were no children classified as preoccupied in this sample. For further analyses, children with dismissing and disorganised attachment were combined into the subgroup of insecure children. Additionally, in line with our main focus of the study, we further analyzed the child’s attachment with the mother. Maternal reflective function ranged from 2 to 8 (M = 4.45, SD = 1.28) which indicates a moderate level of overall RF. Analysis showed that the score was normally distributed (0.29 for skewness and 0.18 for kurtosis).

Preliminary analyses were conducted to explore the relation of demographic factors with the dependent variable–child attachment classification (secure vs insecure). The descriptive data and relations between attachment classification with mother and demographic variables are presented in Table 1. A series of one-way analyses of variance (ANOVAs) and chi-square tests were computed. As shown in Table 1, only age approached statistical significance such that those rated as insecure with mother were younger. The only data not presented in the table is the insignificant χ^2^ analysis examining the relationship of attachment classification and family status (χ^2 =^ 0.92, *p* > 0.05). These findings suggest that the assignment to secure vs insecure attachment classifications was not related to the demographic factors.

To examine the relation of maternal reflective function and the aforementioned variables, correlation analyses were conducted. They did not reveal significant associations, except the tendency to approach significance in relation to maternal RF with her age (r = 0.23, *p* = 0.08).

### 3.2. Association of Attachment, Maternal Reflective Function, and Child’s Gender

Next, to examine the association of maternal reflective function and different child attachment variables, multiple correlations between attachment classifications and CAI subscales evaluations in the total sample were computed first. In order to analyze the association between the maternal RF and their children’s attachment while considering their gender, correlations between maternal RF and all above-mentioned attachment variables were conducted in girls’ and boys’ groups separately. These analyses are presented in Table 2. Further, to consider the possible gender differences, two additional tests were applied. First, group comparison (using t-tests and Mann-Whitney U tests for normally and not normally distributed attachment variables, respectively) revealed no significant differences between boys and girls in CAI subscales and maternal RF. Second, correlations between attachment variables and maternal RF in gender groups were compared using Fisher r-to-Z transformation (see Table 2, last column).

### 3.3. Moderation Analysis

Finally, moderation analysis to evaluate the role of gender in the association between maternal RF and a child’s attachment was accomplished. Results from the PROCESS [34] moderation analyses are displayed in Table 3. The primary focus in the PROCESS moderation model is the coefficient for the product of the independent variable (i.e., maternal RF) and the moderator (i.e., child gender) while accounting for the identified covariates (i.e., child’s age). Results of the final model for attachment security with mother (Nagelkerke R^2^ = 0.27) indicate a significant maternal RF by gender interaction effect (B = 1.18, *p* < 0.05). For a significant interaction, PROCESS then provides the conditional effects of the independent variable at each value of the moderator. As displayed in Table 3 and Figure 1, these conditional effects indicated that higher maternal RF evidenced a higher probability for girls to have secure attachment classification (B = 1.18, *p* < 0.05). Maternal RF was not related to the secure attachment classification of their male child (B = 0.00, *p* > 0.05).

## 4. Discussion

The goal of the present study was to examine the associations between maternal reflective functioning and children’s attachment security in middle childhood. A special focus of this investigation was directed towards the gender-related aspects of this association, i.e., evaluation of possible moderating effects of gender in the relation to mothers’ reflective functioning abilities and attachment security in boys and girls at this age.

The findings of the study demonstrated several interesting aspects of this link. First, overall study results show that maternal RF in middle childhood was positively and significantly linked to the security of their children for the whole sample. Maternal reflective functioning (RF) is considered to be an important factor contributing to children’s attachment security in early childhood [37] and influencing the development of attachment representations in adolescence [38]. Linking data obtained from two sets of narratives, i.e., maternal reflective functioning assessed on the Parent Development Interview Revised (PDI-R2, [34]) and children attachment security accessed by the Child Attachment Interview (CAI, [35]), our study provides new evidence of the associations between mothers’ RF and middle-childhood age children’s attachment considering their gender. Evaluations of this link from a dimensional perspective demonstrated that maternal RF was significantly and positively associated with higher scores in several CAI subscales indicating higher security of attachment relationships for girls but not for boys. Higher maternal RF positively correlated with emotional openness, ability to provide relevant and elaborate examples of experience in relationships, better recognition and integration of positive and negative aspects of parental figures, better ability to describe constructive resolutions of conflicts, higher importance of attachment figures and relationships and higher overall coherence of reflection of attachment relationships in girls. Finally, deeper and more specific analyses demonstrate that relation between child attachment security and maternal RF is moderated by gender. Higher maternal RF indicated a higher probability for girls’ secure attachment but was not related to the secure attachment in the boys’ group.

These findings allow us to draw attention to several aspects that might be looked for in the context of security of attachment in middle childhood. Maternal RF was positively linked to children’s security for the total sample. Borelli and colleagues [22] found similar relationships in their study where specifically child-related reflective functioning of mothers was linked with child attachment security. One aspect of the results in their study also hinted towards importance of gender in this association: being a girl and a higher mother’s child-focused RF were associated with greater attachment security.

Our study adds to these findings, establishing and confirming the importance of gender in this relation. Although numbers of secure children in both groups–girls and boys–did not significantly differ in our study, the association between maternal RF and child attachment security was significant for girls only. Thus, an intriguing question arises: why maternal reflective functioning increases the possibility of secure attachment for girls, but not for the boys, and accordingly, what possibly may account for boys’ security at this age, then? Several points may be highlighted and proposed in this account. One of them is that the security of the child continues from earlier experiences in young childhood and stems from previously developed relationships between mothers and their sons. Some longitudinal studies do confirm this line of interpretation [23]. Structural equation modeling analyses of this study revealed that the overall history of responsive care was meaningfully associated with Security, Avoidance, and Disorganization at age 10 in both mother-child and father-child relationships and that most recent care uniquely predicted security [23]. It should be noted however that assessment of relationship qualities in this study were behaviorally coded. Thus, this requires cautiousness in perceiving our narratively-based data in such a context.

Another possible interpretation of current findings is that patterns of socialization of emotions and conversational style in the family may account for some differences in girls’ and boys’ attachment security indices later. Some studies on dialogues on the past and present experience between young children and their mothers seemed to be of particular interest in understanding the results of our study. Whereas virtually all mothers (at least in modern societies) reminisce with their young children about the experiences they shared, there are clear and enduring individual differences in these processes. The way mothers’ structure reminiscing about shared past experiences with their preschool children is related to their children’s developing autobiographical memory skills and understanding of self and emotion. More specifically, mothers who engage in highly elaborative reminiscing have children who come to tell more coherent and emotionally expressive autobiographical narratives, and these children also show better understanding of self and are better able to regulate emotion than children of less elaborative mothers [39]. Moreover, previous studies in this field intriguingly reveal that reminiscing style is related to gender: findings indicate that both mothers and fathers are more elaborative with daughters than with sons, especially about emotions [31,32]. Both mothers and fathers reminiscing with daughters mention more specific emotions words, and use a greater variety of these words (e.g., talking about being sad, upset, and distressed with girls, but only sad with boys). Parents are also more likely to talk about possible resolutions to negative emotions with daughters than with sons as well as place emotional experiences in a more social context with daughters than with sons [40]. These circumstances may produce girls’ “interrelationships” vocabulary in more advantageous position which consequently may draw onto the coherence of their narratives about relations and emotions at different stages of development and is consistent with findings of the dimensional analysis of relations between different CAI (e.g., ability to provide elaborate examples of experience in relationships, emotional openness, etc.) of our study.

Other authors in more recent research also confirm that assumption. Girls were reported to be more skilled and elaborated in talking about interpersonal emotional events and the causal link has been associated to parent’s emphasis on emotions in early family dialogues with girls [41]. Di Folco et.al. [3] suggested that patterns of socialization of emotions and conversational style in the family may be influencing coherence in children’s narratives (and consequently security). Ongoing discussion in attachment literature raises issues linked to gender differences in attachment security in middle childhood. Gloger-Tippelt and Kappler [42] in a meta-analytical study found that gender is a relevant predictor of attachment classification in middle childhood. Girls were 1.8 times more likely to present secure and 0.4 times less likely to present disorganized attachment narratives as compared to boys when controlling for risk status and age. Suggesting possible explanations, authors declined a biologically based model by Del Guidice [4], according to which gender differences in insecure strategies are due to hormonal changes, adrenarche, and emerge only after age seven, and consequently state that differences in insecure strategies between boys and girls’ attachments should emerge only after age seven, not earlier. Gloger-Tippelt and Kappler [42] oppose this idea by reporting that in their study across all age groups (from 4.5 to 8.5) girls were classified less frequently as avoidant and more often as ambivalent than boys. The authors also make some implications supporting the probability of different styles of emotional socialization in boys and girls that are in line with interpretations of the result of the present study.

### Limitations and Future Research Directions

Findings of our study provide a new but at the same time relatively confined angle of knowledge on gender role on attachment security in relation to parental RF during the middle childhood. First of all, due to rather strict evaluation requirements for narrative collection and evaluation requirements, the investigated sample was rather restricted. Thus, extending the number of participants in future research would be considerably worthwhile and might reveal a richer and more exhaustive picture of the investigated association.

Borelli et al. [22] highlights the question of whether the association between parental RF and child attachment security varies as a function of parent and child sex and if this association might be a function of developmental stage, noting that the RF of the gender-matched parent may become increasingly important to the child as the child nears adolescence and undergoes sex-specific physical changes [43]. On the one hand, procedurally this means focusing not only on the storytelling and reports of children and adults in research of attachment representations in the various periods of the lifespan but also to a special emphasis *on the possible dyadic inter-influence* between the features of these narratives [24]. Thus, including not only maternal but also paternal RF into the investigation of boys’ and girls’ (in)security of attachment in middle childhood would clearly be a very valuable point for further studies.

An ampler set of variables in this association would be valuable to add for understanding of the specifics of the link between parental RF and children’s attachment [44]. Definitely, it would be worthwhile to include different samples (e.g., clinical groups) in future research of gender role in the link between the mother’s and child’s attachment quality research [45].

Certain analyses and interpretation of the results in our study have exploratory nature and are based on the assumption about the close connection between mothers’ reflective functioning and her reminiscing style about the emotional experience with their sons and daughters. Thus, primarily it would be worthwhile to look more specifically into possible links between two psychological constructs, maternal reflective functioning and her reminiscing style, with children in general. Secondly, the research on Lithuanian mothers’ reminiscing style with boys and girls may give more refined and detailed picture of gender-related socialization of children’s emotions specifically in our culture. It is suggested that in the content of mother-child emotion talk we can begin to see how culture’s stereotypes of gender-appropriate emotional reactions are integrated into individual children’s social interactions and how these stereotypes begin to shape the ways in which children come to attribute emotions to themselves and others [30]. Evaluators of our research group observed certain differences of Lithuanian children use of emotion words in their CAI attachment narratives. At first sight, it was noticed that Lithuanian children identify less emotional experiences than usually are observed in the standard protocols used for reliability check of CAI. However, these observations are just of tentative nature and it would be considerably worthwhile to address this observation more specifically in future.

Finally, cross sectional findings of our study also must be addressed while interpreting the results as they put in doubt the distinctions between emotional talk in childhood and later ages. The results of our study indicate that security in relationships was positively related to the child’s age. This echoes the results of some recent studies on mother-adolescents’ (15–18 years) emotional co-narration [46], suggesting that in adolescence, both genders equally express subjective perspectives in their emotional narratives. These findings call for the interesting and promising implementation of studies of longitudinal designs in future.

## 5. Conclusions

Our study extends research on gender related middle-childhood attachment security and provides the first known examination of security in boys and girls of this age in association with maternal reflective functioning. Children undergo many changes in their social life (school demands, relationships with peers and parents) and cognitive development during this developmental stage. Thus, communicating their experience through narratives as well as relating these narratives to their mothers RF is an important context for middle-childhood attachment investigation.

Overall, the results of our study show that girls’ but not boys’ secure attachment is related to maternal RF. These findings suggest possible differentiation in the correlates on the pathway of secure attachment between girls and boys during middle childhood and raise new questions about the life time continuity and changes in attachment relationships for further research. Whilst such data are to be gathered in future, the results obtained in the present study might be useful for the programs aimed at improvement of parent-child relationships.

## Figures and Tables

**Figure 1 ijerph-18-11261-f001:**
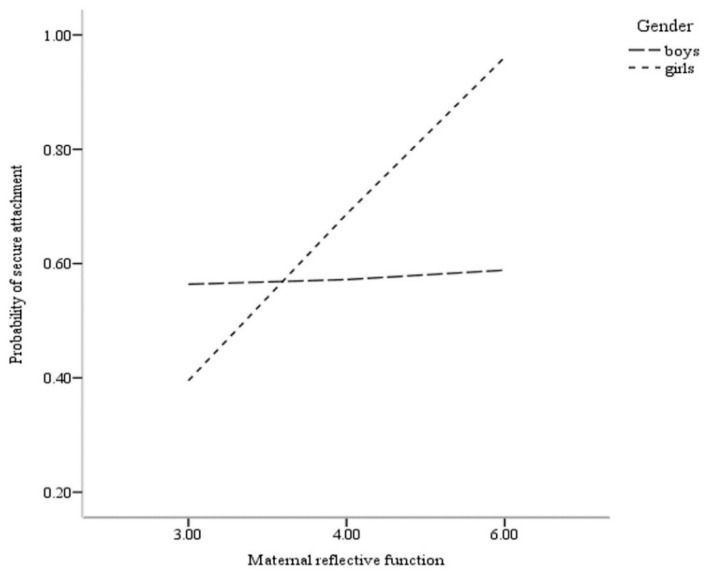
Probability of child attachment security with mother as function of maternal reflective function and gender.

**Table 1 ijerph-18-11261-t001:** Relations between children attachment security and parental demographic variables.

Variable	Attachment to Mother		
	Secure	Insecure	Statistic
Mean age (SD)	8.80 (0.98)	8.22 (1.20)	F(1, 62) = 4.48 *
No. of boys (%)	57.6%	42.4%	χ^2^(1, 64) = 1.25
No. of girls (%)	71.0%	29.0%	
Mother’s age	38.68 (6.51)	36.82 (7.27)	F(1, 60) = 1.06
Mother’s education	3.85 (1.50)	3.57 (1.50)	F(1, 60) = 0.51
Father’s age	39.73 (5.34)	38.81 (5.06)	F(1, 44) = 0.32
Father’s education	3.45 (1.24)	3.33 (1.53)	F(1, 62) = 0.08

*Note.* * *p* < 0.05

**Table 2 ijerph-18-11261-t002:** Relationship between maternal RF and child attachment variables.

Variable		Maternal RF			
	M (SD)	Total Sample	Boys	Girls	Fisher Z-Transformation
*Attachment classification*					
Attachment security with mother ^a^		0.282 *^a^	0.034	0.543 **	−2.186 *
*CAI subscales*					
Emotional openness ^a^	6.20 (2.08)	0.312 *	0.203	0.432 *	−0.976
Balance ^a^	5.92 (2.22)	0.296 *	0.103	0.432 *	−1.366
Use of examples	5.73 (2.00)	0.382 **	0.164	0.586 **	−1.926 *
Anger with mother ^a^	1.57 (1.27)	−0.108	0.040	−0.277	1.235
Anger with father ^a^	1.43 (1.30)	0.018	0.367	−0.230	2.167 *
Idealization of mother ^a^	2.58 (1.67)	−0.118	0.026	−0.259	1.108
Idealization of father	2.50 (1.50)	−0.226	−0.279	−0.204	−0.279
Dismissal of mother ^a^	1.97 (1.58)	−0.195	0.012	−0.442 *	1.852 *
Dismissal of father ^a^	1.77 (1.47)	−0.194	0.044	−0.444 *	1.824 *
Conflict resolution	5.70 (2.28)	0.309 **	0.062	0.589 **	−2.060 *
Coherence	5.65 (2.15)	0.384 **	0.145	0.616 ***	−1.920 *

*Note*. ^a^ Spearman correlation. * *p* < 0.05; ** *p* < 0.01; *** *p* < 0.001.

**Table 3 ijerph-18-11261-t003:** Regression and follow-up analyses predicting child attachment classification (secure vs. insecure) with mother.

Variable	Coefficient	SE	Z	*p*	95% CI
*Regression model*					
Constant	−3.218	2.597	−1.239	0.215	−8.309–1.871
Child age	0.417	0.270	1.546	0.122	−0.111–0.947
Child gender	−4.308	2.451	−1.757	0.078	−9.112–0.495
Maternal RF	0.001	0.311	0.005	0.995	−0.609–0.612
Maternal RF × child gender	1.176	0.581	2.022	0.043	0.036–2.317
*Follow-up analyses*					
Girls	1.178	0.491	0.005	0.016	0.215–2.141
Boys	0.001	0.311	2.399	0.998	−0.609–0.612

*Note.* Child attachment was dummy coded as insecure (0) or secure (1).

## Data Availability

The datasets analyzed in this study are not publicly available due to privacy or ethical restrictions. Data may be available on request from the last author (dr. Rasa Barkauskienė).

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
