# Peer review of "Maternal Reflective Functioning and Attachment Security in Girls and Boys: A Closer Look into the Middle Childhood"

_ijerph, 2021, doi:10.3390/ijerph182111261_

Round 1

Reviewer 1 Report

Thank you for the opportunity to review this paper entitled “Maternal reflective functioning and attachment security in girls 2 and boys: a closer look into the middle childhood”. 

The paper examines the association between maternal RF and child attachment security and whether this link is moderated by gender in 64 mother-child (6-11 years old) dyads from the community sample using attachment-based interviews.

The manuscript contains new information and I appreciated reading it. The methodology is appropriate for the research questions.

In the introduction, the literature review could be more tailored to the focus of the survey stressing further the role of insecure attachment in middle childhood (e.g. Bizzi & Pace, 2020). The results and discussion are presented appropriately. The conclusion may be expanded stressing further the implications of the results.

There is some small refuse (e.g., check the paragraph’s characters). Tables and English may be improved.

In conclusion, the paper addresses the aims and the scope of the Journal selected and it could be published if minor revised.

Author Response

The paper examines the association between maternal RF and child attachment security and whether this link is moderated by gender in 64 mother-child (6-11 years old) dyads from the community sample using attachment-based interviews.

  1. The manuscript contains new information and I appreciated reading it. The methodology is appropriate for the research questions.

       Thank you for your time taken to read our paper and for provided suggestions for its improvement.

  1. In the introduction, the literature review could be more tailored to the focus of the survey stressing further the role of insecure attachment in middle childhood (e.g. Bizzi & Pace, 2020).

We appreciate your comment and suggested reference (Bizzi & Pace, 2020) for tailoring the Introduction part by focusing it to the anlysis of insecure attachment in middle childhood.  We addressed the article by Bizzi and Pace (2020). In this regard we would like to note, that the main goal of the present study was to investigate the role of gender in the link between child’s attachment quality (security and insecurity) and maternal reflective functioning in middle childhood.  Attachment insecurity per se is a little beyond the scope of scope of presented analysis. Definitely it would be worthwhile to include different samples (e.g. clinical groups) in future research of gender role in the link between mother’s RF and child’s attachment quality in future research. We included such consideration into the part of the Discussion (section Limitations and future research).

  1. The results and discussion are presented appropriately.

-

  1. The conclusion may be expanded stressing further the implications of the results.

Thank you for the comment.  Idea about possible practical implications of the findings of our research was added to the Conclusion section.

  1. There is some small refuse (e.g., check the paragraph’s characters). Tables and English may be improved.

Additional proofreading of the text and correcting English and paragraph characters throughout all manuscript text was carried out. Table design was tailored.  

  1. In conclusion, the paper addresses the aims and the scope of the Journal selected and it could be published if minor revised.

Once again, thank you for your time while reading and commenting our manuscript!

Reviewer 2 Report

The aim of the study was to investigate the association between maternal reflective functioning (RF) and child attachment security and whether this link is moderated by gender. The authors address a research topic that is important for better understanding child social and emotional development and thus very important for clinical practice. 

The important strengths of the study is: (i) the use of structured interviews to assess both maternal RF and attachment security that provide much more reliable assessment of these constructs than self-report scales; (ii) focus on child attachment in middle childhood (this topic is understudied); (iii) investigation of gender differences. The findings, which  revealed different patterns of association for key study variables for girls and boys, are very interesting and inspiring for further research. I read the paper with great interest. 

What follows are my reactions and suggestions as I read through the manuscript:

  1. Most parts of the Introduction could be streamlined at least a little bit. However, it would be worth elaborating more on the topic of possible gender differences in relations between maternal RF and attachment security in the Introduction. It would be worth to refer already in this section to the studies that are mentioned in the Discussion eg. studies showing that parents are more elaborative with daughters than with sons, especially about emotions; (ii) parents reminiscing with daughters mention more specific emotions words, and use a greater variety of these words; (iii) parents are more likely to talk about possible resolutions to negative emotions with daughters than with sons as well as place emotional experiences in a more social context with daughters than with sons. Mentioning these studies in the Introduction would provide stronger rationale for investigating gender moderation. Also, maybe based on these findings Authors could consider cautiously putting forward hypotheses that relationships between maternal RF and attachment security could be stronger in girls than in boys?  
  2. The hypotheses should be clearly stated in the “Current Study” section. Authors should more clearly state that some of the analyses have exploratory character.
  3. It would be worth to present results of group comparisons investigating differences between boys and girls in CAI subscales and maternal RF in the table. These findings seem to be important in the context of the topic of the manuscript.
  4. It would be worth discussing at least shortly if cultural factors (e.g. traditional gender roles) could contribute to gender differences in associations between maternal RF and attachment security.
  5. It would be worth to elaborate more on future directions. The findings are so inspiring and interesting that it would be worth to think more about various factors that could be investigated in further research to better understand gender differences in relations between maternal RF and attachment security e.g. what factors should be taken into account (e.g. quality and characteristic of child-parent talk/discussions about mental states), what other maternal and paternal characteristics or specifics of child-parent relationship could play role in development of attachment security in boys?

I thank the authors for the opportunity to review their work, and I hope these comments are helpful to the authors as they pursue this important line of work. 

Author Response

I thank the authors for the opportunity to review their work, and I hope these comments are helpful to the authors as they pursue this important line of work. 

The aim of the study was to investigate the association between maternal reflective functioning (RF) and child attachment security and whether this link is moderated by gender. The authors address a research topic that is important for better understanding child social and emotional development and thus very important for clinical practice. 

The important strengths of the study is: (i) the use of structured interviews to assess both maternal RF and attachment security that provide much more reliable assessment of these constructs than self-report scales; (ii) focus on child attachment in middle childhood (this topic is understudied); (iii) investigation of gender differences. The findings, which revealed different patterns of association for key study variables for girls and boys, are very interesting and inspiring for further research. I read the paper with great interest. 

Thank you for taking time to read our manuscript and for you insightful comments for its improvement. Below we present the information about corrections that has been carried out according to these comments.

What follows are my reactions and suggestions as I read through the manuscript:

  1. Most parts of the Introduction could be streamlined at least a little bit. However, it would be worth elaborating more on the topic of possible gender differences in relations between maternal RF and attachment security in the Introduction. It would be worth to refer already in this section to the studies that are mentioned in the Discussion eg. studies showing that parents are more elaborative with daughters than with sons, especially about emotions; (ii) parents reminiscing with daughters mention more specific emotions words, and use a greater variety of these words; (iii) parents are more likely to talk about possible resolutions to negative emotions with daughters than with sons as well as place emotional experiences in a more social context with daughters than with sons. Mentioning these studies in the Introduction would provide stronger rationale for investigating gender moderation. Also, maybe based on these findings Authors could consider cautiously putting forward hypotheses that relationships between maternal RF and attachment security could be stronger in girls than in boys?  

All sections of Introduction part were carefully revised and tailored by adding some more data from the field of the research on mother-child communication about emotions. The idea the idea about the differences in mother’s elaborateness and expressivity while talking about the emotional experience with daughters and sons was clarified and stressed. We suggested that these differences may manifest in the link between maternal reflective functioning and attachment quality with boys and girls.

       2. The hypotheses should be clearly stated in the “Current Study” section. Authors should more clearly state that some of the analyses have exploratory character.

We refined a clarified the hypotheses in the Current study section of Introduction part suggesting  the greater possibility for maternal reflective function to be associated with attachment security in girls as follows:

Presented disputes in literature on empirical findings as well as suggested in this paper overlap between the concept of mother’s elaborative style in communication with children about emotions and the construct of maternal reflective functioning al-low us to assume the greater possibility for maternal reflective function to be associated with attachment security in girls.

    3. It would be worth to present results of group comparisons investigating differences between boys and girls in CAI subscales and maternal RF in the table. These findings seem to be important in the context of the topic of the manuscript.

Analyses on comparisons between the groups of boys and girls CAI subscales showed no significant differences. This information is presented in the manuscript Results part (section 3.2) : group comparison (using t-tests and Mann-Whitney U tests for normally and not normally distributed attachment variables, respectively) revealed no significant differences between boys and girls in CAI subscales and maternal RF. Therefore, decision to not include this information into the table was made.

    4. It would be worth discussing at least shortly if cultural factors (e.g. traditional gender roles) could contribute to gender differences in associations between maternal RF and attachment security.

Thank you for this useful comment. We did add some reflections on the idea of culturally determined   gender-related patterns of parent language about emotions with boys and boys and girl into the part of the Discussion (section  Limitations and Future research directions): It is suggested that in the content of mother-child emotion talk we can begin to see how culture’s stereotypes of gender-appropriate emotional reactions are integrated into individual children’s social interactions and how these stereotypes begin to shape the ways in which children come to attribute emotions to themselves and others [30].

    5. It would be worth to elaborate more on future directions. The findings are so inspiring and interesting that it would be worth to think more about various factors that could be investigated in further research to better understand gender differences in relations between maternal RF and attachment security e.g. what factors should be taken into account (e.g. quality and characteristic of child-parent talk/discussions about mental states), what other maternal and paternal characteristics or specifics of child-parent relationship could play role in development of attachment security in boys?

The Discussion section on Limitations and Future Research Directions was enlarged by adding more reflections and ideas about possible directions in future investigations of the link between parental reflective functioning and children’s’ attachment qualities.    

I thank the authors for the opportunity to review their work, and I hope these comments are helpful to the authors as they pursue this important line of work. 

Once again, thank you for your time and valuable insights! We hope that changes made by following your suggestions improved the manuscript and made it appropriate for publication.